# A Live Cell Protein Complementation Assay for ORFeome-Wide Probing of Human HOX Interactomes

**DOI:** 10.3390/cells12010200

**Published:** 2023-01-03

**Authors:** Yunlong Jia, Jonathan Reboulet, Benjamin Gillet, Sandrine Hughes, Christelle Forcet, Violaine Tribollet, Nawal Hajj Sleiman, Cindy Kundlacz, Jean-Marc Vanacker, Françoise Bleicher, Samir Merabet

**Affiliations:** 1IGFL, CNRS UMR5242, ENS-Lyon, UCBL-1, INRA USC1370, 32 Av. Tony Garnier, 69007 Lyon, France; 2Department of Developmental and Cell Biology, University of California, Irvine, CA 92697, USA; 3LiPiCs, 46 Allée d’Italie, 69007 Lyon, France

**Keywords:** Cell-PCA, BiFC, ORFeome-wide, human HOX proteins, protein-protein interactions, live cells

## Abstract

Biological pathways rely on the formation of intricate protein interaction networks called interactomes. Getting a comprehensive map of interactomes implies the development of tools that allow one to capture transient and low-affinity protein–protein interactions (PPIs) in live conditions. Here we presented an experimental strategy: the Cell-PCA (cell-based protein complementation assay), which was based on bimolecular fluorescence complementation (BiFC) for ORFeome-wide screening of proteins that interact with different bait proteins in the same live cell context, by combining high-throughput sequencing method. The specificity and sensitivity of the Cell-PCA was established by using a wild-type and a single-amino-acid-mutated HOXA9 protein, and the approach was subsequently applied to seven additional human HOX proteins. These proof-of-concept experiments revealed novel molecular properties of HOX interactomes and led to the identification of a novel cofactor of HOXB13 that promoted its proliferative activity in a cancer cell context. Taken together, our work demonstrated that the Cell-PCA was pertinent for revealing and, importantly, comparing the interactomes of different or highly related bait proteins in the same cell context.

## 1. Introduction

Organismal development and fitness depend, for a large part, on their cell protein content. Proteins are versatile molecules that work in a crowded environment, establishing a number of interactions with other surrounding proteins. These protein–protein interactions (PPIs) form intricate networks called interactomes which change from cell to cell and stage to stage. 

Characterizing these dynamic molecular networks is a key issue in understanding protein function and implies the development of highly sensitive tools. One of the main experimental approaches for capturing PPIs is the yeast two-hybrid system (Y2H), which relies on the indirect readout of a reporter gene to reveal any interaction between a bait protein and its candidate partners [1]. Although very popular, Y2H presents the major inconvenience of being performed in a heterogeneous context (i.e., in yeast for non-yeast proteins and in the context of proteins fused to heterologous DNA-binding and activation domains). 

Recent approaches based on biotin-ligase enzymes and the liquid chromatography-mass spectrometry (LC-MS) identification of biotinylated partners constitute promising alternatives for the characterization of the interactomes in specific cell and tissue types [2,3]. These approaches are appropriate for the identification of endogenous interactomes but are not compatible with the more systematic and high-throughput interrogations of binary PPIs with dedicated candidate libraries. 

To date, the use of fluorescent [4] and non-fluorescent [5] reporters in protein-fragment complementation assays (PCAs) represents the simplest and most sensitive tools for screening PPIs in live cell conditions. These approaches rely on the properties of the reporters to be reconstituted from separate N- and C-terminal fragments upon spatial proximity (Figure 1A). However, all currently described fluorescent PCA-based screens rely on open reading frames’ libraries (ORFeomes), which are transiently used in the cell either upon the co-transfection (Co-PCA strategy: Appendix A; [6]) or transduction of lentiviral particles (Re-PCA strategy: Appendix A; [7,8,9]. These strategies do not allow comparisons of the interactomes of different bait proteins since the libraries are not stably conserved and are systematically screened in one shot with no replicates in the same cell population. 

Here, we proposed an alternative cell line PCA-based experimental strategy (Cell-PCA) that relied on the establishment of a cell line expressing a PCA-compatible human ORFeome (Figure 1B). This cell line could be amplified and used several times to simultaneously screen for the interactions of different bait proteins, eventually allowing comparisons of the interactome properties against the same ORFeome in the same cell context (Figure 1B).

As a proof of concept, we applied this novel experimental strategy to the HOX protein family, which is involved in the regulation of numerous processes during embryonic development [10,11] and adult life [12]. HOX proteins are transcription factors (TFs) and, therefore, act by regulating the expression of downstream target genes in vivo. Several cofactors have been identified for different individual HOX proteins in various cell and developmental contexts [3,13,14,15], but no systematic large-scale interaction screening has been performed on several HOX members in the same biological system. As a consequence, very little is known about their general and specific interactome properties. For example, the question of HOX cofactor specificity remains poorly understood: is there a large proportion of specific versus common cofactors between different HOX proteins? Along the same line, work with mouse HOXA1 showed that a number of cofactors were not traditional TFs, suggesting that HOX proteins could act at different regulatory levels [13]. Whether this property could apply more largely to other HOX protein members remains to be investigated. 

To tackle the issue of HOX interactome properties, we presented the first large-scale screening of the PPIs of eight different human HOX proteins in the same cell population. Our results showed that TFs were generally not employed as HOX-specific cofactors but were instead used in specific combinations in the different HOX interactomes underlying common biological functions. In contrast, we observed that HOX proteins had a general propensity to interact with non-TFs and that these interactions were more HOX-specific than the interactions with TFs. Several of these interactions were also individually validated through BiFC and co-immunoprecipitation experiments. Finally, we revealed a novel interaction that was important for HOXB13 proliferative activities in a prostate-cancer-derived cell line. 

Taken together, our results established the Cell-PCA as an innovative and promising experimental strategy for assessing the issue of interactome specificity in a live cell context. 

## 2. Results

### 2.1. Cell-PCA Screen Design

The fluorescence-based complementation approach, also called BiFC (bimolecular fluorescence complementation), relied on the properties of the hemi-fragments of monomeric fluorescent proteins, such as the GFP (green fluorescent protein) or Venus, to reconstitute a functional fluorescent protein upon spatial proximity [4]. For the Cell-PCA design, we used a library of 8200 ORFs and fused this set of ORFs to the C-terminal fragment of the blue fluorescent protein Cerulean at the 5′ end (fragment CC, Figure 1C and Materials and Methods). This fragment could complement the N-terminal fragment of Venus (VN, leading to a Venus-like fluorescent signal) or Cerulean (CN, leading to a Cerulean-like fluorescent signal), enabling one to simultaneously visualize the interactions of two different bait proteins with a common cofactor [15,16,17]. In addition, the small size of the C-terminal fragment (82 residues long) made it more neutral when compared to the N-terminal fragment (173 residues long) for fusion protein constructs [18].

The CC-ORF library was cloned in a lentivirus vector downstream of the regulatory sequences (*Tet-Responsive Element* (*TRE)*, Figure 1C and Appendix A) that respond to the tTA (tetracycline-controlled transactivator) factor in the presence of doxycycline. The CC-ORF pooled plasmid library was used to produce lentiviral particles and was used in the subsequent infection of HEK-293T cells (see Materials and Methods). Referring to the functional titer, the pooled lentiviral libraries were transduced at a low multiplicity of infection (MOI) to achieve only one stably integrated CC-ORFeome in most cells (see Materials and Methods). This operation was repeated two times, and the two resulting cell lines were named CC-HEK-1 and CC-HEK-2, which, respectively, encompassed the 5745 and 5973 genes (Figure 1C and Appendix A). A total of 5005 genes were found to be present in the CC-HEK1 and CC-HEK2 cell lines (Appendix A), and 1328 CC-ORFs were not present at a significant frequency (less than 10 counts) in the CC-HEK cell lines (Appendix A). The basal expression of the CC-ORFeome library was further verified in the established CC-HEK cell lines through immunostaining against the CC fragment (Appendix A). 

Each VN-bait fusion protein was under the control of the constitutive *CMV* promoter, and the experimental conditions of transfection to obtain comparable expression levels between the different VN-bait fusion proteins had previously been established in HEK293T cells (see Materials and Methods and [16,19]). Transfecting the VN-fusion plasmid into the established CC-HEK cell line would lead to BiFC-positive signals only when an interaction occurred between the VN-bait and the CC-prey proteins produced from the corresponding integrated CC-ORF (Figure 1D). The fluorescent cells were then sorted using flow cytometry from which the genomic DNA was extracted to prepare a sequencing library with the specific oligonucleotides that matched the CC-ORF construct (Figure 1D, Appendix A). The presence and relative abundance of the integrated CC-ORFs in the sorted fluorescent cells were assessed using a dedicated next-generation sequencing (NGS) approach (see Materials and Methods). This targeting approach allowed the reduction of the sequencing effort to read and investigate only the beginning of the inserted CC-ORF instead of throughout the complete genome or at the insert fragments (which are highly variable in size). This strategy improved the sequencing coverage together with a reduced cost.

### 2.2. Proof-of-Concept Cell-PCA Screen for HOXA9 Interactomes

As a proof-of-concept, we performed a high-throughput interaction screen on the human HOXA9 protein, whose interaction with the two known cofactors PBX1 and MEIS1 has been extensively described using BiFC in the HEK293T and other cell lines [19]. In particular, this previous work established that the VN-HOXA9 fusion topology was appropriate for deciphering the properties of HOX interacting properties with PBX1 and MEIS1 in vitro and in live cells [19]. We, therefore, decided to use the same VN-HOX9 fusion protein to perform the large-scale BiFC interaction screen. We also considered a mutant form of HOXA9 as a supplementary control in the experimental design (VN-HOXA9^W^). This form is mutated on a unique conserved Trp residue that mediates the interaction with the PBX cofactor in the context of the HOX/PBX dimeric complexes [19,20]. This conserved Trp residue was also shown to have additional and versatile activities that changed depending on the cell context, suggesting that it could interact with other cofactors [19]. This Trp-mutated form of HOXA9 was, therefore, considered to be a good control for assessing the specificity and sensitivity of our experimental tools. 

The pilot screens with VN-HOXA9 and VN-HOXA9^W^ were sequentially performed in a single replicate in each CC-HEK-1 and CC-HEK-2 cell line for two main reasons. First, as previously mentioned, transduction was performed with a low MOI to get only one CC-ORF construct in most of the cells (see Materials and Methods). This transduction condition resulted in the incomplete integration of the CC-ORFeome library in each cell line (around 70%). Performing the screen in the two CC-HEK cell lines increased the proportion of the human ORFeomes that could be considered (reaching 82%). Second, we wanted to assess whether the strategy of considering interactions that were either specific to one CC-HEK cell line or common to the two CC-HEK cell lines could be pertinent. Given that a majority of the commonly integrated CC-ORFs in the two CC-HEK cell lines displayed a similar range of integration frequencies (Appendix A, see also discussion), the CC-HEK cell lines were considered to be biological replicates that could be used to assess the level of the reproducibility of positive interactions (Figure 2A). Interactions that were specific to a cell line or common to the two cell lines were selected by applying different threshold criteria (see Materials and Methods).

The basal expression of the TRE promoter was used in the BiFC screen since it was sufficient to reveal the expression of the integrated CC-ORFs (Appendix A). In addition, to simplify the protocol, having a minimum expression level for each integrated CC-ORF also allowed the screen to be performed in more stringent conditions without the overexpression of the corresponding prey protein. Under these conditions, fluorescent signals were only observed upon the co-transfection of VN-HOX9 (Appendix A), and this pattern was systematically obtained in the subsequent screens. 

According to our experimental protocol and selection criteria, we found a total of 413 (6% of the integrated CC-fusion ORFeome) positive interactions among which 115 were common to the two CC-HEK cell lines (Figure 2B and Appendix A). These candidates were selected after applying a log2-fold change enrichment threshold that was different depending on the positive interaction status of one or both CC-HEK cell lines (Figure 2C and Materials and Methods). To assess whether our post-NGS selection criteria were correct for unique or duplicated positive interactions, we randomly selected 17 CC-ORFs with a broad range of enrichment scores that were positive either in one (7 CC-ORFs) or both CC-HEK cell lines (10 CC-ORFs). The interaction with PBX1, which was not present in our CC-ORF libraries, was used as a positive control. We also considered one CC-ORF (C6orf201) that was negative in the screen. These candidates were tested by doing individual BiFCs with VN-HOXA9, using the interaction with PBX1 as a calibrator of the imaging parameters in order to evaluate the statistical significance across several biological replicates (see Materials and Methods). Individual interactions were also tested by performing co-immunoprecipitation experiments with HA-tagged HOXA9, and therefore validated through an orthogonal experimental approach. Results showed that the interaction status was confirmed for all tested candidates (Figure 2D and Appendix A). Interestingly, BiFC analyses revealed various interaction profiles in the live cells (Figure 2D and Appendix A). Collectively, these observations confirmed that the applied filtering criteria were appropriate for selecting positive interactions when considering one or both cell lines. 

We next performed the ORFeome-wide BiFC screen on the two CC-HEK cell lines transfected with VN-HOXA9^W^ as the reference condition for future comparisons.. This mutated form of HOXA9 had fewer positive interactions than wild-type HOXA9 (342 in total, corresponding to 5% of the integrated CC-fusion ORFeome and using identical selection criteria as those for HOXA9; Figure 3A and Appendix A). A total of 96/413 (23%) HOXA9 interactions were also found with HOXA9^W^, indicating that our tools were sensitive enough to reveal the different interactomes upon a single amino acid mutation (Figure 3B). The observation that 246 interactions were specific to HOXA9^W^ also highlight that the Trp mutation induced more of a gain than a loss in the HOXA9 interaction potential. 

The comparison of the biological functions that were enriched in the HOXA9 and HOXA9^W^ interactomes revealed that the classical HOX functions related to morphogenesis- and DNA-binding-dependent transcription were present in both interactomes although they had different levels of enrichment (highlighted in blue and green in Figure 3C). In contrast, several functions related to epigenetic and chromatin organization were lost in the HOXA9^W^ interactome (highlighted in violet in Figure 3D). In addition, the Trp mutation affected the activity of HOXA9 in muscle formation (highlighted in light orange in Figure 3D) and led to a loss of several functions involved in cell division and cancer progression (highlighted in red in Figure 3D). This last effect has previously been reported in several studies with HOXA9 [21,22]. Finally, the ectopic functions revealed with the Trp mutation were also linked to the cell division processes (Figure 3D), suggesting that this residue could have dual activities. Other ectopic functions were related to more specific terms, such as the CREB1, YAP/TAZ and ATAC complexes (Figure 3D). These results recalled previous observations with *Drosophila* Hox proteins, which have been shown to establish ectopic interactions and to perform additional functions when mutated in the same Trp-containing motif [14,23].

Together, the results obtained with VN-HOXA9 and VN-HOXA9^W^ validated the proof-of-concept ORFeome-wide interaction screen used to capture (sensitivity) and distinguish (specificity) the interactomes of two highly related HOX proteins. These results were encouraging for applying this experimental strategy to a more systematic HOX interactome screen exploration. We, therefore, applied the Cell-PCA strategy to capture the interactomes of seven additional human HOX proteins, tackling the general issue of human HOX interactome specificity in the same cell system. 

### 2.3. Using Cell-PCA for a Global Comparison of HOX Interactomes 

HOX members belonging to anterior (HOXA1 and HOXA2), central (HOXC6, HOXA7 and HOXC8) and posterior (HOXA9, HOXD10 and HOXB13) paralog groups were chosen for the ORFeome-wide comparison of different HOX interactomes (Figure 4A). HOX proteins were fused to the VN fragment at their N-terminus since this fusion topology had previously been described to be compatible for deciphering the interaction properties of a protein interacting with PBX1 and MEIS1 in vitro and in live cells [16,19]. Each VN-HOX encoding plasmid was transfected in the two CC-HEK-1 and CC-HEK-2 cell lines for the ORFeome-wide BiFC screen. Transfection conditions for homogenous and comparable expression levels have previously been established [16,19]. We applied the same selection criteria as previously described for HOXA9 screening (e.g., sorted fluorescent cell, candidate interaction partners, etc.) (see Materials and Methods).

Results showed that each HOX member had a variable number of positive interactions (between 4.9% and 6.7% were positive interactions; Figure 4B and Appendix A). As expected, interactions with TFs were enriched (40% of all HOX interactions, 595/1491) although TFs represented 20% of the tested CC-ORFs (1342/6713). The majority of interactions were not unique, being found with one or more additional HOX proteins (between 64% and 83%; Figure 4C). Still, each HOX protein showed a specific cluster of interactions (between 17% and 36%; Figure 4C). Interestingly, the HOX-specific clusters were poorly enriched with DNA-binding-domain (DBD)-containing TFs (between 9% and 26%; Figure 4D). In contrast, DBD-containing TFs were significantly more enriched in the non-specific HOX fractions (between 41% and 58%; Figure 4D). 

As expected, the heatmap of the top-20 enriched functions of all the HOX interactions revealed functions linked to transcriptional regulation, chromatin organization, cell differentiation and tissue/organ morphogenesis (Figure 4E). Surprisingly, prostate cancer was also specifically enriched in several HOX proteins although it corresponded to a deregulated biological function. (Figure 4E). Even more surprisingly, this function was most enriched in HOXA2 although prostate cancer is more often associated with the deregulated activity of the posterior HOX members (see below). However, recent work showed that HOXA2 is associated with aggressive prostate cancer, underlining the robustness of our data [24,25,26].

The analysis of the interactome involved in mRNA transcription confirmed that the majority of the interactions established between HOX proteins and TFs were not specific (Appendix A). Instead, it was the combination of the full set of interactions established by each Hox protein with their TFs that was specific (Figure 5). This observation suggested that HOX transcriptional specificity resulted principally from specific combinations of interactions rather than from interactions with specific TFs. 

Altogether, the results obtained showed that the Cell-PCA approach was efficient for revealing the specific interactomes of eight different HOX proteins in the same cell context. 

### 2.4. CREB3L4 Interacted with HOXB13 and Promoted Its Proliferative Activity in a Prostate Cancer Cell Context

Prostate cancer was among the top-20 enriched biological functions revealed upon the analysis of HOX interactomes. The global involvement of HOX proteins in human cancer is well established, and they can have pro- or anti-tumoral activities depending on the HOX protein and the cancer type [27,28]. Particularly, several HOX proteins have been described to promote or inhibit prostate cancer progression [29,30]. These studies relied on the analysis of the HOX expression level in primary prostate cancer cells and functional readouts in established prostate-cancer-derived cell lines. Along this line, one of the best case studies for prostate cancer is HOXB13, which has been described in several instances to be both overexpressed and required for prostate cancer cell proliferation and metastasis [31,32]. We, therefore, looked more precisely at the HOXB13 interactome involved in prostate cancer and found several interesting candidates known to display the same pro-oncogenic activity (Figure 6A). 

Several model cell lines derived from prostate cancers have been established. In particular, the role of HOXB13 has been well studied in the PC-3 cell line, and it has been found to have a role in promoting proliferation and malignancy [33,34,35]. We hypothesized that this proliferative activity could depend on its interaction with the cofactors that were revealed in our screening. In such a scenario, the loss of the candidate cofactor should affect the proliferative activity of HOXB13. To test this hypothesis, we selected *CREB3L4,* which has been described to be expressed in several prostate-cancer-derived cell lines, including PC-3 cells [36]. Interestingly, CREB3L4 and HOXB13 were predicted to interact when using AlphaFold (Figure 6B–B’’) [37,38]. The role of this candidate cofactor in HOXB13 proliferative activity was tested using the xCELLigence system, which uses cellular impedance to continuously measure the number, size and surface attachment strength of adherent tumor cells [39,40,41]. We first confirmed that *HOXB13* and *CREB3L4* were expressed in PC-3 cells and that siRNAs were efficient in affecting the corresponding endogenous gene expression (Appendix A). The effects of siRNAs were analyzed 24 h, 48 h, 72 h and 96 h post-transfection. The effect of each individual siRNA against *HOXB13* or *CREB3L4* was first apparent at 72 h. These effects were moderate, similar for each siRNA and more pronounced at 96 h post-transfection (Figure 6C). Importantly, combining two siRNAs against *HOXB13* and *CREB3L4* led to a significantly more pronounced effect on cellular impedance than that of each single siRNA, which was already seen at 24 h post-transfection (Figure 6C). Given that the use of siRNAs against *CREB3L4* did not affect the expression of *HOXB13* (Appendix A), we concluded that HOXB13 and CREB3L4 could work as a cooperative dimeric protein complex that could promote the proliferation of PC-3 cells. The BiFC and Co-IP experiments confirmed that the two proteins could indeed interact and form a protein complex (Figure 6D,E). 

Altogether, these results showed that CREB3L4 could work as the collaborative partner of HOXB13, potentially through direct protein–protein interactions, to promote its proliferative activity in prostate-cancer-derived PC-3 cells.

## 3. Discussion

### 3.1. Cell-PCA Allowed Interactomes to Be Captured and Compared in the Same Live Cell Context

Protein interactomes are versatile networks involving hundreds of transient and low-affinity interactions. Over the last years, several experimental strategies based on PCA systems have been developed to capture these molecular interactions, leading to promising alternatives in addition to LC-MS or Y2H-based approaches. In this context, the BiFC-based PCA is particularly well adapted for revealing pair-wise interactions in live cells and has been applied to several screening strategies to study the interactomes of various bait proteins [4,6,7,42]. Although these screens were based on an off/on readout with no enrichment scores in any replicate, they revealed specific sets of interactions that were further confirmed with alternative molecular and functional assays. Altogether, this previous work established BiFC as a powerful method for revealing novel interactomes. Our work further enriched the repertoire of the applicability of BiFC for large-scale protein interaction screens, in particular, by proposing an experimental setup that allowed the use of the same cell line when performing different screens. This strategy provided an additional level of information that allowed the interactomes of different bait proteins to be compared. 

The Cell-PCA relies on the establishment of a cell line that integrates a BiFC-compatible ORFeome. This cell line can be amplified and used several times for BiFC interaction screens with different bait proteins. In principle, this strategy can be applied in any cell line of interest as long as it is easily transfectable, which facilitates maximizing the number of VN-bait expressing cells. This parameter is important when considering that only a small proportion of these expressing cells will be positive for the cell sorting and subsequent NGS analysis. As a proof of concept, we used the HOX protein family and two different CC-HEK cell lines, and we proposed stringent filtering parameters to select the most relevant candidate interaction partners. The screen was voluntarily performed in conditions of low expression levels for each CC-ORF prey construct, allowing the acquisition of specific BiFC signals with the transfected VN-bait protein (itself under multiple copies). 

The two CC-HEK cell lines were used as biological replicates by considering the common pool of randomly integrated CC-ORFs. We also noticed that a higher number of positive interactions were systematically revealed in the CC-HEK-1 cell line when compared to the CC-HEK-2 cell line. This was probably due to better transfection conditions in the CC-HEK-1 cell line (which is an inherent part of variability when considering two different biological replicates). Still, we found a relevant proportion of reproducibility of positive interactions when considering the common pool of integrated genes in the two CC-HEK cell lines with our stringent selection criterion (log2FC ≥ 1.5), which ranged from 20% to 40% depending on the HOX protein (Appendix A). This score was in the range of the reproducibility rate described for approaches based on high-throughput mass-spectrometry protein complex identification (around 19% when considering the proteins present in the two datasets [43]) or Y2H (around 20% [44]). This proportion was lower, however, than the proportion of shared ORFs, which were integrated (5005/6713, i.e., 75%), suggesting that there could be a proportion of false negatives. This proportion could be explained by our stringent log2FC criteria, which aimed at getting positive candidates with a high confidence. It could also be explained by the fact that each CC-ORF was randomly inserted at a variable frequency in the genome (Appendix A). Along this line, we noticed an inverse correlation between the number of common positive CC-ORFs and the variation of the insertion frequency score (based on the number of counts) between the two CC-HEK cell lines (Appendix A). For example, the proportion of common positive ORFs between CC-HEK1 and CC-HEK2 reached 73% for HOXA1 when considering ORFs that varied less than three times between the two cell lines (Appendix A). This point suggested that the level of reproducibility would probably be higher between replicates performed with the same CC-HEK cell line. Using a CRISPR-based system to target a unique genomic insertion site in all the CC-ORF constructs could constitute an interesting alternative in the future with this regard [45]., 2019). Nevertheless, the overall number of positive interactions found in our screens was higher than the number of interactions obtained with other experimental approaches (see below), suggesting that the potential number of false negatives was not a strong limitation. On the contrary, given the high sensitivity of BiFC, it is particularly important to apply a high selection criterion to limit the number of potential false-positive interactions. 

The specificity of the tools was confirmed with the results obtained for HOXA9 and the mutated HOXA9^W^ construct, with a small proportion of common interactions (23% of the HOXA9 interactions were also found in HOXA9^W^). Along the same line, we obtained only 15 interactions that were common to all the tested HOX proteins (all were nuclear proteins, and most are DBD-containing TFs; Appendix A), which confirmed that each sorted BiFC-positive cell was the result of specific interactions with the HOX protein and not with the VN tag. Nevertheless, having a neutral VN bait protein could also help in selecting the most relevant interactions rather than the non-specific background interactions, especially when testing few bait proteins or when in the absence of a negative control. It could be the VN tag alone or one fused to a fluorescent protein such as mCherry. 

Finally, the positive interactions that were randomly picked up from the screen for HOXA9 were subsequently validated by performing individual BiFC and Co-IP experiments. Co-IPs were performed independently of the complementation system, and they systematically reproduced the BiFC result. Similar observations have been reached in previous BiFC screens of cells [46] 12/23/22 5:38:00 PM or of the *Drosophila* embryo [14], underlining that BiFC and Co-IP can be used as complementary approaches for validating an interaction potential between two artificially expressed candidate proteins. Our results also validated our stringent selection criteria as the appropriate criteria for selecting relevant candidate interactions. 

The Cell-PCA approach did not reveal all the expected or previously described interactions. For example, we found that not all HOX proteins were able to interact with the representatives of the generic PBX and MEIS cofactors present in the two CC-HEK cell lines (PBX3 and MEIS2). PBX3 was captured for HOXC6 and HOXA7, and MEIS2 was captured for HOXA1 and HOXC8. Our stringent selection criteria eliminated HOXA1, HOXA2, HOXA9, HOXC10 and HOXB13 for PBX3, and it eliminated HOXC6, HOXA9, HOXD10 and HOXB13 for MEIS2 (a 0 read was obtained in the two CC-HEK cell lines for HOXA2 and HOXA7 with MEIS2). These results suggested that the fusion topology might not be the most appropriate in several cases for revealing the interactions with BiFC. Performing a second screen with alternative fusion topologies could help resolve this issue. 

Although further developments could be done in the future, our results showed that the Cell-PCA was a sensitive, specific and robust approach for performing ORFeome-wide interaction screens upon simple transfection of a bait protein. The Cell-PCA not only simplified the protocol (the screen could be performed in a classical A2 laboratory environment since it did not rely on systematic transduction as previously described [7]) but also enabled the testing of different bait proteins in the same batch of cells, therefore providing a unique level of information for comparative interactome analyses. 

### 3.2. HOX Interactomes Were Revealed with Cell-PCA: Molecular Properties and Comparison with Existing Databases

The analysis of eight different HOX interactomes revealed several unexpected and interesting molecular features. For example, there was an important proportion of non-TFs in the overall pool of HOX interacting proteins. Similar observations were seen in a Y2H screen of HOXA1 [13,47]. Along the same line, the Hox protein Ultrabithorax (Ubx) has been shown to establish tissue-specific interactions with the cofactors involved in translational regulation [3]. Altogether, this novel layer of interactions illustrated the ability of HOX proteins to be engaged in the regulation of several post-transcriptional regulatory processes, a level that remains poorly investigated to date.

Interestingly, TFs were mostly found in non-HOX specific interactions, whereas the proportion of non-TFs was enriched in HOX-specific interactions. Accordingly, HOX interactomes related to transcriptional regulatory processes contained distinct combinations of a majority of non-specific interacting TFs as exemplified in the enriched RNA-Pol-II function (Figure 5 and S7). This observation underlined that HOX transcriptional specificity mostly relied on the establishment of specific combinations of interactions with TFs that had the potential to interact with several HOX proteins. This molecular mode of action has already been proposed for *Drosophila* Hox interactomes, suggesting that it could be a general and conserved feature underlying Hox transcriptional specificity [14]. 

We also compared our positive interactions to the publicly available Biogrid database (https://thebiogrid.org/, accessed on 13 July 2020), which compiles all the characterized protein–protein interactions with genetic and/or molecular evidence in different model systems. In particular, Biogrid includes the interactions listed in Bioplex (https://bioplex.hms.harvard.edu/, accessed on 13 July 2020). Bioplex profiled the interactions obtained from C-terminally FLAG-HA-tagged baits within the human ORFeome v8.1 when performing Co-IP on different human cell lines, including HEK293T cells, followed by the LC-MS identification of endogenous binding partners. 

In general, we found between 5% and 7% of the 6713 integrated genes to be positive HOX interaction candidates in our BiFC screens. In comparison, a Y2H screen performed with HOXA1 against the human ORF v3.1 revealed only 59 positive interactions, or less than 1% of the screened bait proteins [13]. This result illustrated one major drawback of the Y2H heterologous system: the lack of sensitivity and number of false negatives generally obtained in Y2H screens [44]. Nevertheless, considering all the positive HOXA1 interactions listed in Biogrid from different Y2H screens revealed a total number of 347 positive interactions, or around 3.5% of the total human ORFeome. The Cell-PCA revealed 5.4% positive interactions for HOXA1 (363/6713), which was in the same order of magnitude. The higher proportion of positive interactions found with the Cell-PCA could be explained by it having a more appropriate (with the natural DNA-binding of HOXA1 in a human live cell context) and, therefore, more sensitive biological system than Y2H. Still, we found that 71 proteins out of the 254 captured interactions in the Y2H screens, were present in our CC-HEK cell lines and identified as HOXA1-interacting candidates in our BiFC screen, showing a highly relevant percentage of overlap (28%; Appendix A). In contrast, only 5/347 Y2H HOXA1 positive interactions are also listed in Bioplex. It is interesting to note that 2/5 of these listed Bioplex cofactors were present in our CC-HEK cell lines, and that one of them has been captured with VN-HOXA1 (ZNF503). In any case, it is important to stress that, in contrast to the cell-type specific Co-IP of endogenous cofactors, Y2H- or BiFC-based approaches are revealing a global potential for discovering the interactions between bait and candidate interaction partners. 

The comparison of the interactions listed in Biogrid for HOXC8 showed the same range of overlap: among the 119 listed cofactors of HOXC8, 98 were present in our CC-HEK cell lines, and 29 were positive (29.6%; Appendix A). Far fewer cofactors were described for the other HOX proteins in Biogrid, which limits the scope of the conclusions. For example, 20/27 of the listed cofactors of HOXA9 were present in our CC-HEK cell lines, and 7/20 were positive in our BiFC screens, showing a similar range of positive percentages (35%, Appendix A). Overall, these observations underlined that the interactions revealed with the Cell-PCA contributed a significant proportion of the current interactomics database (which resulted mostly from Y2H screens). 

In conclusion, our work confirmed that HOX proteins had a strong potential to engage in a number of interactions with various partners, especially those particular ones, which were poorly investigated in post-transcriptional regulatory processes. Our understanding of their specific molecular mode of action certainly requires a better consideration of these supplementary levels of regulation in the appropriate cell or tissue systems in the future. 

## 4. Materials and Methods

### 4.1. Cell Lines

HEK-293T and PC-3 cells were purchased from European Collection of Authenticated Cell Cultures (ECACC) through the biological resource center Anira-AGC platform of SFR Biosciences UAR3444/US8, Lyon. Both cell lines were cultured in Dulbecco’s modified Eagle’s medium (DMEM-GlutaMAX-I, Gibco by Life Technologies) and were supplemented with 10% (*v/v*) heat-inactivated fetal bovine serum (FBS) and 1% (*v/v*) penicillin–streptomycin (5000 U penicillin and 5 mg streptomycin/mL), and they were incubated at 37 °C in an atmosphere of 5% CO_2_. HEK-293T-CC-ORFs were cultured as above with 0.3 µg/mL of puromycin (Gibco, Cat No. A1113803) in their culture medium.

### 4.2. Plasmids

The bait plasmids, pcDNA3-VN-HOXs (expressing VN-HOXs), were made as described previously [16,19]. The lentiviral pLV-CC-ORF vector collection was kindly provided by P. Mangeot (CIRI, ENSL, Lyon, France). The genomic DNA was used for library preparation and was subjected to sequencing by in-house Ion Proton NGS sequencing system (PSI, IGFL, Lyon, France). About 8200 ORFs from the V3.1 version of the hORFeome were fused at the 5′ end to the C-terminal part of the mCerulean gene (encoding the last 155-238 aa) using Gateway^®^ technology.

For individual BiFC tests, constructs were cloned from the pLIX _403 vector (a gift from David Root; Addgene plasmid # 41395; http://n2t.net/addgene:41395, accessed on 13 July 2020; RRID:Addgene_41395). 

DNA sequencing of all constructions was carried out by GENEWIZ Company (Germany). All vectors are available upon request.

### 4.3. Lentivirus Preparation and Infection

The pooled lentiviral constructs, pLV-CC-ORFs, were packaged into lentivirus particles at the AniRA-Vectorology core facility (SFR Biosciences UAR3444/US8, Lyon, France). HEK-293T cells were transduced in independent replicates with two batches of lentivirus (CC-ORF library 1 and CC-ORF library 2) at a low multiplicity of infection (0.3) to approximately achieve one-gene-one-cell condition [48] with ≥500X representation. Culture medium was supplemented with 8 µg/mL polybrene (TR-1003, Sigma-Aldrich, Lyon, France) at the time of transduction and was changed the next day. Two days after transduction, cells were selected with 0.5 µg/mL puromycin (Cat No. A1113803, Gibco/Thermo Fisher, France) for 4 days until the uninfected control cells completely died and the selected cells reached near confluency. Final amplified transduced cells were split into aliquots of 4 × 10^6^ cells each and were stored in liquid nitrogen for future screens. 

### 4.4. Immunostaining of CC-ORF Transduced HEK Cells

1 × 10^5^ cells were seeded on glass coverslips in 24-well plates. Twenty-four hours after plating, cells were fixed in 4% paraformaldehyde in PBS for 15 min at room temperature, were permeabilized in 0.3% Triton X-100 for 10 min and were rinsed in PBS. The cells were preincubated in 3% bovine serum albumin–PBS at room temperature for 1 h and were incubated overnight at 4 °C with primary antibodies against CC-ORF (1:1000, A11122, rabbit anti-GFP, Invitrogen, France). They were then incubated with the corresponding fluorescein-conjugated secondary antibodies (1:1000, A11008, Alexa Fluor 488 anti-rabbit, Invitrogen, France) for 2 h at room temperature. Coverslips were mounted in VECTASHIELD Antifade Mounting Medium with DAPI (Cat No. LS-J1033-10, Vector laboratories, Burlingame, CA). Images were acquired with a Zeiss LSM780 confocal microscope (Carl Zeiss, Jena, Germany).

### 4.5. Cell-PCA Screen

8.10^6^ CC-HEK cells (~ 800X representation) were thawed and passaged for 2 population doublings for recovery. For each screen, aliquots of 6 × 10^6^ recovered cells were seeded on two 6-well plates (500 k cells/well) and were grown for 24 h to achieve a final confluence of around 80%. The basal expression of the TRE promoter was used for the expression of the CC-ORFs in the cell population. The transfection of different pcDNA3-VN-HOX bait plasmids was performed using the jetPRIME reagent (Ref 114-15, Polyplus Transfection, France) following manufacturer’s instructions. Transfection conditions have previously been established for comparable expression levels (Dard et al., 2019b, 2018b). After 18 h of transfection, all transfected cells were pooled, and 15000 BiFC-positive cells were subsequently sorted using a BD FACS Aria II cell sorter (performed at AniRA-Cytometry core facility of the SFR Biosciences UAR3444/US8, Lyon, France). After each screen, sorted cells were harvested, and genomic DNA was extracted using PureLink Genomic DNA Mini Kit (Cat No. K182001, Invitrogen, France) according to manufacturer’s instructions. The genomic DNA was used for library preparation and was subjected to next-generation sequencing by in-house sequencing platform (PSI, IGFL, Lyon, France).

### 4.6. Next Generation Sequencing and Identification of the Positive hORFs 

Libraries were constructed using our own proprietary protocol design in order to enrich sequences covering the beginning of all the hORFs that were inserted in the genome (see details in Appendix A). After a size selection using SPRI beads to meet Ion Torrent requirements, the qualified and quantified barcoded libraries were multiplexed in an equimolar manner and were sequenced with the Ion Proton Sequencer using a P1 chip following the manufacturer’s recommendations.

NGS raw data were analyzed with the Galaxy instance [49] of the ENS of Lyon and were maintained by the Centre Blaise Pascal (CBP, ENSL, France). A dedicated Galaxy pipeline was created to identify the hORFs detected through sequencing and the associated read counts of each barcoded sample. After demultiplexing, the raw reads were trimmed to remove very low-quality bases at the 3′ and 5′ ends using a sliding window process. Through construction, the libraries were oriented, and all the reads began with the same short plasmid sequence that is present upstream of any inserted hORF. This sequence was removed by Cutadapt [50], with a maximum allowed error rate of 0.15. Only the trimmed reads beginning with ATG were retained for further analysis. This last step removed any reads that could result from a non-specific PCR amplification. Because read length is variable with the Ion Torrent technology, the reads were then trimmed to 50 bp to ensure that they all had the same length. These reads were next compared to the hORFeome v3.1 database with BLAST tool [51] using strict conditions (only one hit retained at least 98% of identity on 95% of the query coverage; matches starting at position 1 of the hORFs). For each hORF that obtained hits, the number of reads matching this hORF was counted and summed up in a table for further analyses. According to the criteria used, only hORFs that had more than a 3-base difference in their first 50 bp could be differentiated (98% of the genes of the hORFeome could still be distinguished; Appendix A). Moreover, when the beginnings of the hORFs were identical or nearly identical (mainly hORFs corresponding to different isoforms of the same gene), the read was assigned to only one of the possible alternatives. Since the analyses were done at the gene level, the counts of ORFs for the same gene were added up.

### 4.7. Identification of HOX-Interacting ORF Candidates

After raw data cleaning, the sequencing counts of each gene in each library were normalized to 10 M. To limit artefacts, count tables were denoised using an arbitrary threshold of at least 50 counts/10 M reads in the control cell libraries. For the sorted cells, a higher threshold of 500 counts (x 10) was applied considering that they constituted an enriched population when compared to non-sorted/cold cells. Then, the number of counts in sorted cells was divided by the number of counts in the cell-controlled libraries, and the log2FC was calculated as the enrichment score (ES) for each gene. 

To generate the final list of HOX-interacting candidates, genes that were present in both replicates with an ES >= 1.5 were combined, assigning the highest ES obtained to each gene. To get an extensive view of all potential interactions, we also considered the top enriched genes (with an ES ≥ 6) of interactions that were present in a single replicate. 

### 4.8. Individual BiFC Validation in Live Cells 

For transfection, 3.10^5^ cells were seeded on glass coverslips in 6-well plates and incubated for 24 h. Then, cells were transfected with jetPRIME (Ref 114-15, Polyplus Transfection, France) following manufacturer’s instructions. A total of 1.75 µg of plasmid DNA was transfected per well: 750 ng of plix-VN-HOXA9, 750 ng of plix-CC-ORF and 250 ng of plix-mCherry plasmids. After 18 h of incubation in the presence of doxycycline (100 ng/mL final), the cell-coated coverslip was taken and mounted carefully on a glass slide for image capturing under Zeiss LSM780 confocal microscope (Carl Zeiss, Jena, Germany). All samples were imaged using identical settings and were quantified as previously described [16]. Two biological replicates were systematically performed using the interaction of HOXA9 and PBX1 as a positive BiFC control and using the mCherry reporter to assess transfection efficiency.

### 4.9. Individual Co-Immunoprecipitation (IP) Validation 

For Co-IP assays, HEK-293T cells were plated at 2 million cells in a 10 cm petri dish and were transfected with pLIX_403-2HA-HOXA9 (4µg) and/or pLIX_403-CC-bait (4 µg) and with PEI at a ratio of N/P = 5. Cells were returned in a complete medium and were supplemented with 200 ng/mL of doxycycline to induce expression of the bait and the prey. Cells were harvested 48 h post-transfection in phosphate-buffered saline (PBS), and pellets were resuspended in NP40 buffer (20 mM Tris pH 7.5, 150 mM NaCl, 2 mM EDTA, 1% NP40) and were treated with Benzonase (E8263-5KU, Sigma-Aldrich, Lyon, France). Anti-HA magnetic beads (#88836, Pierce, Thermo Fisher Scientific, France) were added to the protein extract, were incubated for 2 h and were washed five times with NP40 buffer. All samples were resuspended in Laemmli buffer for immunoblotting analysis. All buffers were supplemented with protease inhibitor cocktail (P8340, Sigma-Aldrich, Lyon, France), 1 mM DTT and 0.1 mM PMSF. Input fractions represented 1–10% of the immunoprecipitated fraction. 

For Western blot analysis, proteins were resolved on 12% SDS-PAGE, were blotted onto PVDF membrane (ISEQ00010, Millipore, USA) and were probed with specific antibodies after saturation. The antibodies (and their dilution) used in this study were anti- Histone H3 (rabbit) (1:10000, ab 1791, Abcam, Paris, France), anti-GFP (rabbit) (1:2000, A11122, Invitrogen, France), anti-HA (mouse) (1:3000, 901513, Biolegend, San Diego, CA, USA) and anti-HOXB13 (rabbit) (1:1000, PA5-98698, Invitrogen, France). 

All blots were developed by enhanced chemiluminescence reaction (ECL, GE Healthcare, USA), secondarily coupled with HRP (1:5000, Promega, France). Visulaization of the bands of interest was performed using Amersham ImageQuant 800 (Cytiva, Marlborough, MA, USA).

### 4.10. Functional Enrichment and Interactome Analysis

Both functional and interactome analyses were performed with Metascape (https://metascape.org/, accessed on 13 July 2020) [52] using custom analysis settings. Subsequently, Cytoscape v3.8.2 [53] was conducted to visualize representative HOX functional interactomes.

In functional enrichment analysis, the HOX-interacting protein candidates were searched against Gene Ontology Biological Processes, KEGG pathway, CORUM and Reactome databases. A p-value cutoff ≤ 0.01 was used to determine significant functional terms. They were then hierarchically clustered into a tree based on kappa statistical similarities among their gene memberships. A kappa score ≥ 0.3 was applied as the threshold to cast the tree into term clusters. We selected the term with the best p-value within each cluster as its representative term and displayed them in the heatmaps. 

Physical PPIs from multiple data sources were captured for construction of the interaction networks. Homo sapiens was selected as the organism for subsequent analysis. Min network size of 3 was regarded as cut-off criterion for network visualization, and disconnected nodes were hidden. The complex identification algorithm MCODE [54] was used to identify highly interconnected clusters in the network. The most important protein complex clusters in the PPI network were extracted with default settings of MCODE: degree cutoff = 2, node score cutoff = 0.2, max depth = 100 and k-score = 2. For each complex, it further applied function enrichment analysis and used significantly enriched terms for annotation of their biological roles. Following manual curation, the similar terms were combined and classified into non-redundant parent functions and categories, which were visualized with heatmaps. Based on combined data set, all representative HOX functional interactomes were generated and visualized with Cytoscape.

### 4.11. xCELLigence Assays

The xCELLigence system (ACEA Biosciences Inc., San Diego, CA, USA), which records cellular events in real time by measuring electrical impedance across microelectrodes integrated on the bottom of culture plates (E-plates), was utilized in proliferation experiments. 

First, cell culture media were added to each well of 16-well E-plates (ACEA Biosciences Inc., San Diego, CA, USA) to measure background impedance. Then, 1.5 × 10^5^ of PC3 cells were transfected with a mixture of control, ERRα, HOXB13 and CREB3L4 siRNAs at 8 pmol/mL using INTERFERin (Polyplus Transfection, France) according to the manufacturer’s protocol. Transfected cells were directly seeded on E-plates (2.5 × 10^4^ cells/well), and the impedance was measured every 15 min for 96 h. The impedance signal was proportional to the number of cells proliferating in each well and was displayed as cell index. Data were analyzed with the RTCA Software 2.0 and were presented as mean +/− SEM of three experiments performed in triplicate or quadruplicate.

### 4.12. siRNAs Used in This Study Were from Eurogentec

HOXB13: GUUCAUCACCAAGGACAAG and CUUGUCCUUGGUGAUGAAC

CREB3L4: CCAGUUCUCCUAUGCUCUA and UAGAGCAUAGGAGAACUGG

ERRα: GGCAGAAACCUAUCUCAGGUU and CCUGAGAUAGGUUUCUGCCUC

### 4.13. siRNAs and RT-QPCRs

For siRNA transfection, 3.10^−5^ cells per ml were seeded on 6-well plates, and 25 pmol/mL of total siRNA was transfected with INTERFERin (Polyplus Transfection, France) according to the manufacturer’s recommendations. Total RNAs were extracted using the guanidinium thiocyanate/phenol/chloroform method. A total of 1 µg of RNA was converted to first-strand cDNA using the RevertAid First Strand cDNA Synthesis Kit (Thermo Scientific, MA, USA). Real-time qPCRs were performed in 96-well plates using the IQ SYBR Green Supermix (Catalog # 1708880, BioRad, CA, USA). Data were quantified using ΔΔ-Ct method and were normalized to 36b4 expression. 

### 4.14. Sequences of the Primers Used in This Study

36b4: GTCACTGTGCCAGCCCAGAA and TCAATGGTGCCCCTGGAGAT

HOXB13: CAGATGTGTTGCCAGGGAGA and TGCTGTACGGAATGCGTTTC

CREB3L4: AGCTGCCCTTTGATGCTCAT and CGGTCAGGAACAGGGTTTGA

ERRα: CAAGCGCCTCTGCCTGGTCT and ACTCGATGCTCCCCTGGATG

## Figures and Tables

**Figure 1 cells-12-00200-f001:**
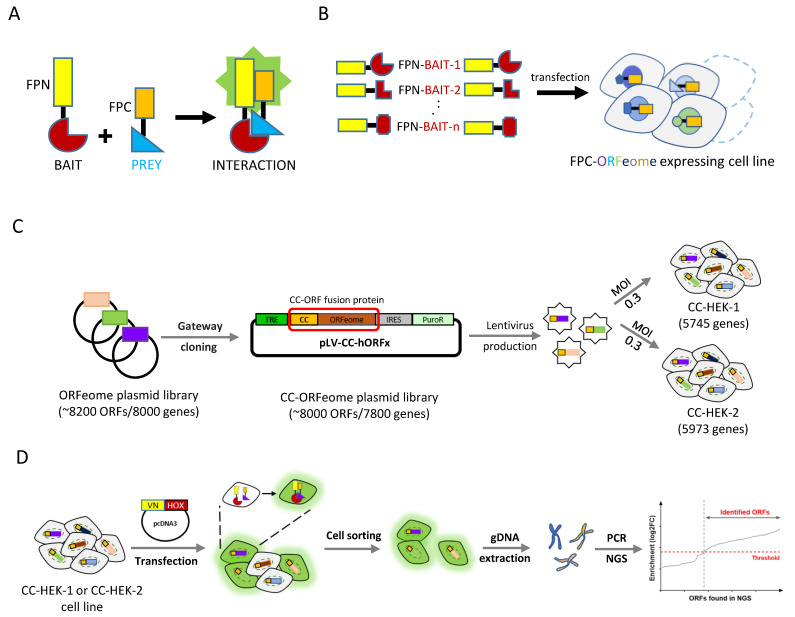
**Principle of the protein complementation assay (PCA) and its applications in large-scale interaction screens.** (**A**) Application of fluorescence-based PCAs to reveal interactions between two candidate partners. The N (FPN)- or C (FPC)-terminal fragment of the fluorescent protein (FP) is fused to one of the two putative interaction partners (bait and prey proteins). The interaction between the bait and the prey proteins allows the reconstitution of the fluorescent protein and the emission of fluorescent signals upon excitation. This principle of complementation has also been developed with enzymes for large-scale interaction screens (see for example [5]). (**B**) Application of PCA-based strategies to large-scale interaction screens in living cells. The cell line PCA-based screening strategy (Cell-PCA) relies on the use of cell lines established with an inserted FPC-fusion library (**D**). These cell lines can be used multiple times in screening for interacting partners of different FPN-fusion bait proteins introduced by transfection. (**C**,**D**) Experimental procedure for the Cell-PCA-based screen. (**C**) A pool of ~ 8200 human ORFs derived from the hORFeome *v3.1* is cloned *en masse* with Gateway^®^ LR reaction into the lentiviral vector pLV-CC (Appendix A), subsequently generating the CC-ORFeome plasmid library (pLV-CC-hORFs). The final expression plasmid library (~ 8000 ORFs) is used to produce lentiviruses and to infect two different batches of HEK293T cells to generate two different cell lines (CC-HEK-1 and CC-HEK-2). (**D**) Each CC-HEK cell line can be transfected with the VN-HOX-encoding plasmid. Any interaction with a CC-ORF leads to fluorescent cells that are collected using flow cytometry. Genomic DNA (gDNA) is extracted from the fluorescent sorted cells, and interacting ORFs are identified through a next generation sequencing (NGS)-dedicated approach. CC: C-terminal fragment of mCerulean (residues 155–238). VN: N-terminal fragment of mVenus (residues 1–172). MOI: multiplicity of infection.

**Figure 2 cells-12-00200-f002:**
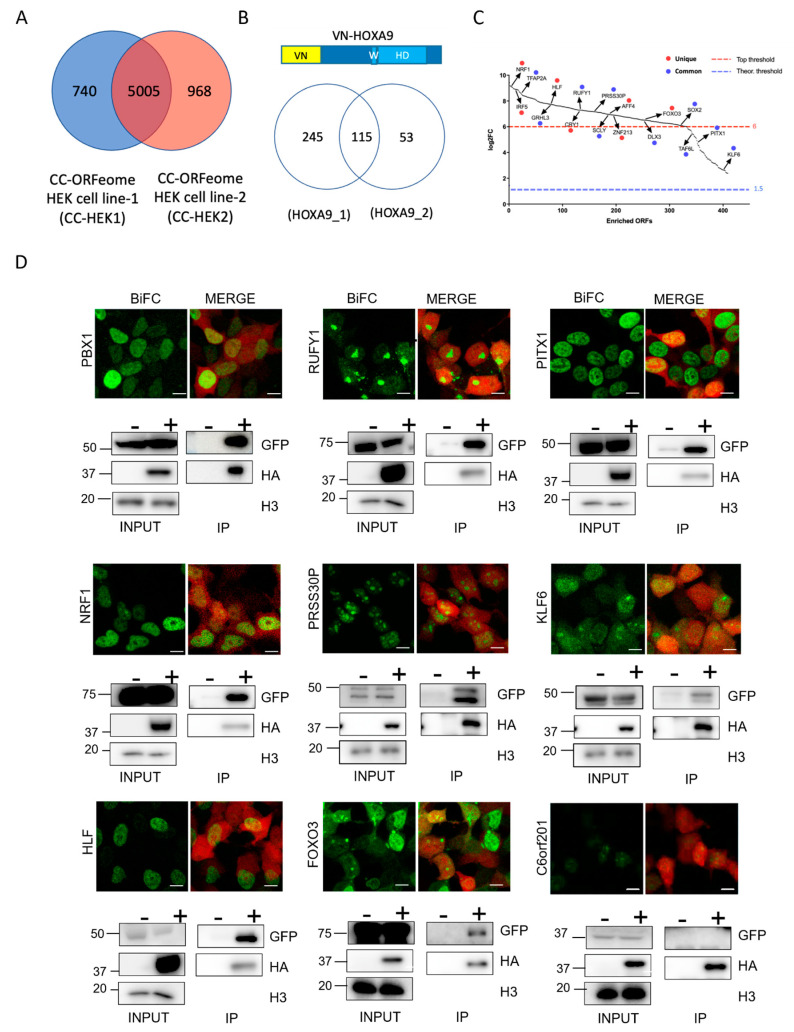
**Establishing the Cell-PCA screening strategy with HOXA9.** (**A**) Venn diagram depicting the number of integrated ORFs in the CC-HEK-1 (blue) and CC-HEK-2 (red) cell lines. (**B**) Venn diagram showing the number of HOXA9-positive ORFs in the two CC-HEK cell lines. VN-HOXA9 is schematized above the Venn diagram (with the Trp-containing motif—W—and the homeodomain—HD). (**C**) Plot of the 413 selected HOXA9-interacting candidates, ranked from the most to the least enriched in the Cell-PCA assay. Among them, 17 CC-ORFs were randomly picked for individual validation with BiFC using two criteria: the 7 red dots were unique to one CC-HEK cell line with a log2-fold change (FC) superior to 6; the 10 blue dots were common and were revealed in the two CC-HEK cell lines with a log2FC that was superior to the background (1,5). See also Materials and Methods. (**D**) Illustrative confocal pictures of BiFC and co-immunoprecipitation (Co-IP) in HEK293T cells of HOXA9 and the selected candidates as indicated. All pictures are illustrative of two independent biological replicates. For BiFC experiments, the mCherry reporter (red, merge panels) was indicative of the transfection efficiency. Note the various intra-cellular BiFC profiles with the different candidates. BiFC with the negative C6orf201 control was also negative in BiFC (with a fluorescent signal below 15% of the fluorescence intensity resulting from HOXA9/PBX1 BiFC on average) and Co-IP experiments. Scale bar: 10 μm. Co-IP was performed with HA-HOXA9, and the CC-ORF was revealed with anti-GFP. When two candidates were of different sizes, the protein extracts were loaded on the same gel (for PITX1/NRF1 and PRSS3OP/KLF6). Staining with Histone H3 antibody validated the correct protein extraction in each condition. “+“ and “−”, respectively, denote the co-transfection or lack of co-transfection of HA-HOXA9 with the CC-ORF construct. The protein size scale is indicated on the left side (KDa).

**Figure 3 cells-12-00200-f003:**
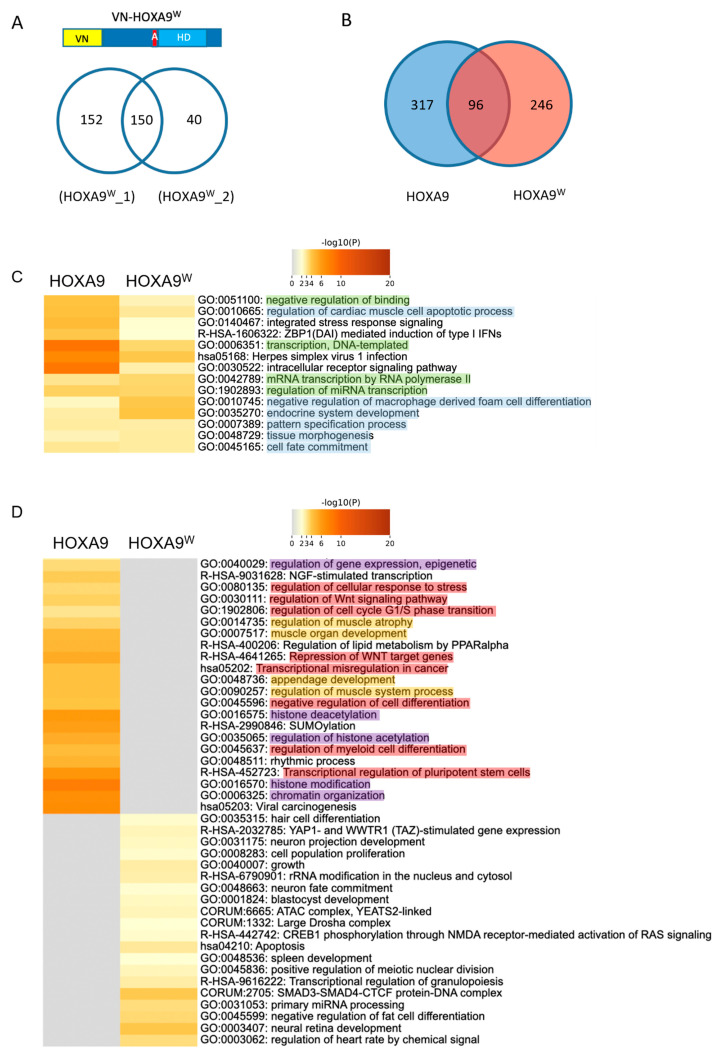
**Cell-PCA revealed distinct interactomes of HOXA9 and HOXA9^W^.** (**A**) Venn diagram of HOXA9^W^-interacting ORFs in the two CC-HEK cell lines. The Trp (W) mutation into an Ala (A) is shown in the schematized VN-HOXA9^W^ protein above the Venn diagram. (**B**) Venn diagram showing the comparison of HOXA9 and HOXA9^W^ interactomes. (**C**) Heatmap of the top-20 enriched biological functions in both HOXA9 and HOXA9^W^ interactomes. One row per function, using a discrete color scale to represent statistical significance (from high (dark red) to no (gray) significance). Blue and green colors highlight functions involved in transcriptional regulation or morphogenesis, respectively. (**D**) Heatmap showing the specific biological functions underlying HOXA9 and HOXA9^W^ interactomes (considering the 317 HOXA9-specific and 246 HOXA9^W^-specific interactions). Violet, orange and red colors, respectively, highlight functions involved in epigenetics, muscle formation and cell proliferation/cancer, which were lost upon the W mutation.

**Figure 4 cells-12-00200-f004:**
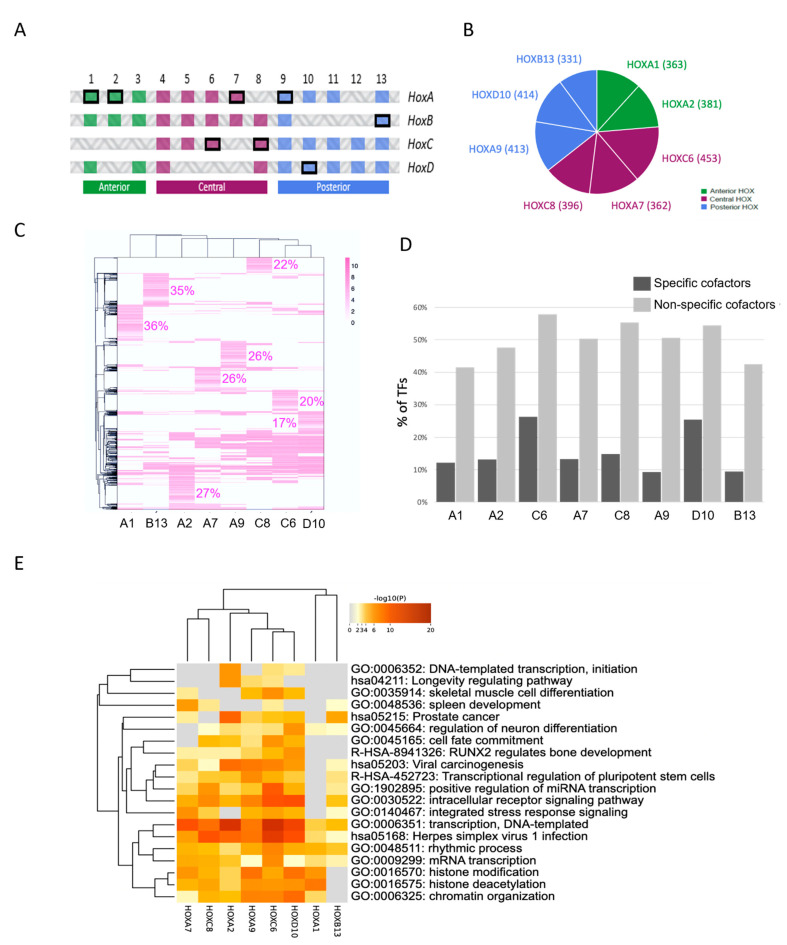
**Application of Cell-PCA for global HOX interactome screening and comparisons.** (**A**) Schematic arrangement of the 39 human HOX genes. HOX genes belonging to anterior (green), central (purple) and posterior (blue) paralog groups. The 8 HOX genes used in the screen are highlighted (framed in black). (**B**) Pie chart illustrating the number of interacting ORFs identified for each HOX protein in the Cell-PCA screens. (**C**) Heatmap of interacting ORFs identified for each HOX protein in the Cell-PCA screens. Hierarchical clustering was performed on both columns and rows according to the Pearson distance based on log2-fold change (FC) values using average method. The proportion of HOX-specific interactions is indicated (as a % of total interactions of each HOX protein). Scale bar indicates enrichment score (log2FC) for each HOX-interacting ORF. The proportion of HOX-specific interactions is indicated. (**D**) Distribution of DNA-binding-domain-containing transcription factors (TFs) of HOX-specific (dark gray) or non-HOX-specific (light gray) interactors. Note that TFs constitute between 50% and 80% of the total interactions, but they are systematically more enriched in the non-HOX-specific category. (**E**) Heatmap of the top-20 enriched functional profiles of the different HOX-interacting proteins. Hypergeometric p-values and enrichment factors were calculated and used for filtering. A hierarchical clustering was performed on both columns and rows based on kappa statistical similarities among their gene memberships. A discrete color scale was used to represent statistical significance. Gray color indicates a lack of significance. The large majority of these functions were linked to gene regulation or cell differentiation and development as expected.

**Figure 5 cells-12-00200-f005:**
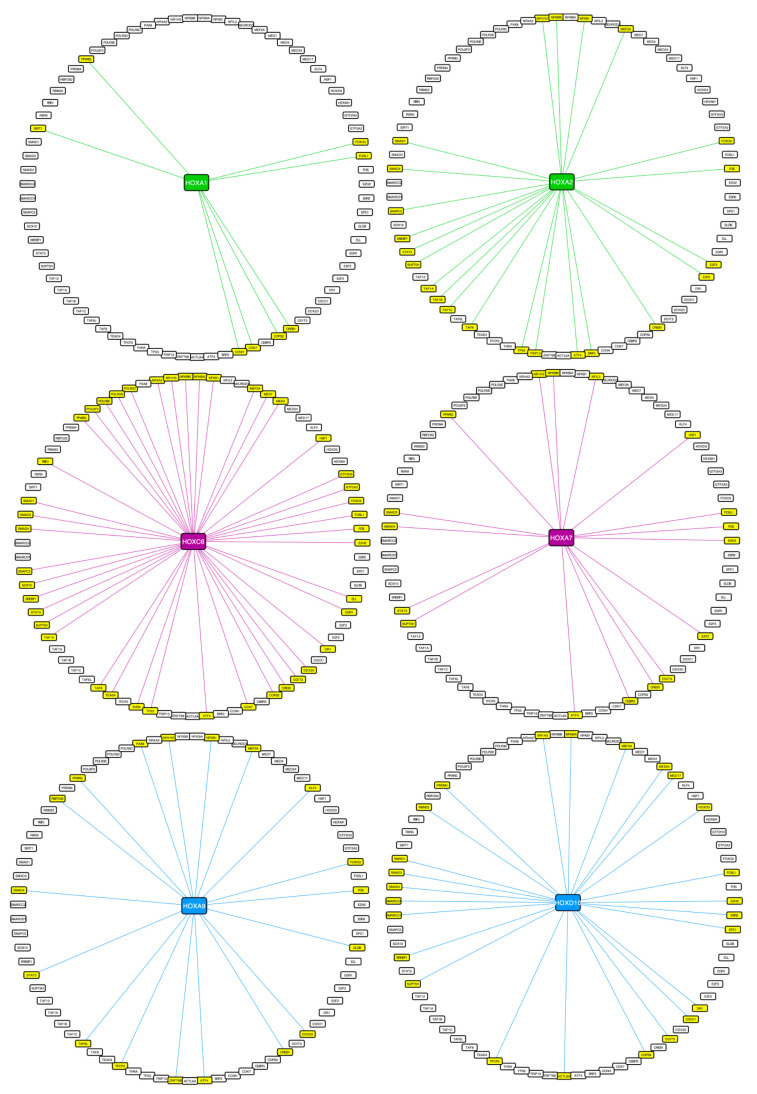
**HOX transcriptional interactomes involved in RNA-Pol-II dependent transcription function that was found to be enriched in HOX BiFC screens.** Interacting TFs are highlighted in yellow and connected with a trait of the HOX proteins in each representative anterior (HOXA1 and HOXA2 (green)), central (HOXC6 and HOXA7 (violet)) and posterior (HOXA9 and HOXD10 (blue)) HOX interactome.

**Figure 6 cells-12-00200-f006:**
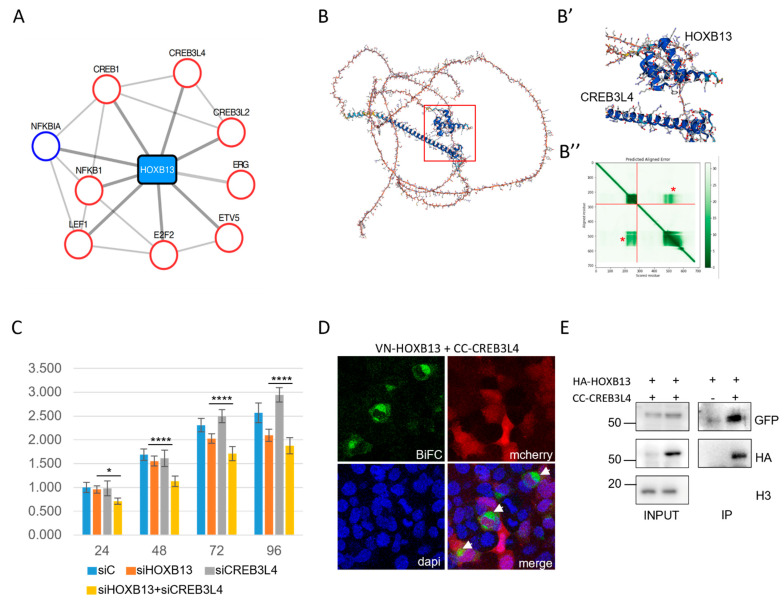
**CREB3L4 is a novel cofactor of HOXB13 that can promote the proliferation of prostate-cancer-derived PC-3 cells.** (**A**) Interactome of the HOXB13 enriched cluster involved in prostate cancer. Candidate cofactors have a known tumor suppression gene (TSG (blue circle)) or oncogenic (red circle) function. (**B**) AlphaFold prediction of the interaction between CREB3L4 and HOXB13. (**B’**) Enlargement of the interaction interfaces between the homeodomain (HD) of HOXB13 and the alpha helix of CREB3L4. (**B”**) Prediction score of intra-domain interactions of HOXB13 (upper dark green box; corresponds to the HD) and CREB3L4 (lower dark green box; corresponds to the alpha helix) and extra-domain interactions between the HD of HOXB13 and the alpha helix of CREB3L4 (green boxes highlighted by a red star). The first 0-280 residues correspond to HOXB13, and the following 280-700 residues correspond to CREB3L4. (**C**) xCELLigence assay of PC-3 cells transfected with the different siRNAs as indicated. siC = siRNA control (see also Materials and Methods). Measures were performed at different time points post-transfection and resulted from three independent biological replicates. Two-way ANOVA with Tukey’s multiple comparisons; * *p* < 0.05 and **** *p* < 0.0001. (**D**) Illustrative confocal picture of BiFC (green) of VN-HOXB13 and CC-CREB3L4 in fixed HEK293T cells. The mCherry reporter (red) stains show transfection and DAPI (blue) indicates cell nuclei. A typical enrichment was observed in dividing nuclei (white arrows). (**E**) Co-IP of HA-tagged HOXB13 and CC-CREB3L4 co-expressed in HEK293T cells. The Co-IP was performed with anti-HA, and CREB3L4 was revealed with anti-GFP, which recognizes the CC fragment (see Materials and Methods). “+” and “−”, respectively, denote the co-transfection or lack of co-transfection of HA-HOXB13 with CC-CREB3L4. The protein size scale is indicated on the left side (KDa). Confocal and Western blot pictures are illustrative of two independent biological replicates.

## Data Availability

Not applicable.

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
