# Peer review of "A Live Cell Protein Complementation Assay for ORFeome-Wide Probing of Human HOX Interactomes"

_cells, 2023, doi:10.3390/cells12010200_

Round 1

Reviewer 1 Report

This manuscript by Yunlong Jia et al. presents a new methodology, they called Cell-PCA, to carry out proteome-wide protein-protein interaction screenings in live animal or human cell models. This assay is basically based on the BiFC concept, but it involves cells stably transfected for wide libraries of ORF-half-Venus fusions, which allows to use libraries for several assays to be run in parallel in order to lead to comparative interactomics in the same live cell context.

This methodology is innovative and clearly presented. The authors also provide convincing proof-of-concept experiments. A key issue in the approach however consists in the specificity and sensitivity of the Cell-PCA approach. The authors indicate that the specificity and sensitivity of Cell-PCA was established by comparing the interactomes obtained with wildtype and mutant HOXA9 proteins, but the outcome of these interaction screenings raise important questions mainly about the sensitivity of the assay. These questions should be addressed or discussed before acceptation of the manuscript for publication.

Major issue

The main question raised by the interpretation of the data presented in the manuscript concerns the sensitivity of the assay, which by essence renders questionable the comparability of the interactomes established by Cell-PCA.

The authors established two similar cell lines, CC-HEK-1 and CC-HEK-2, which were independently infected by two batches of the same library of lentiviral constructs coding for CC-ORFs. The characterization of these cell lines highlighted a high overlap of integrated constructs between the cells: 5005 integrated ORFs are shared by the two cell lines, 740 (CC-HEK-1) and 968 (CC-HEK-2) constructs being unique to either cell line. In contrast, while comparing the positive interactions retrieved with the same protein (HOXA9 WT) only 115 interactions (out a total of 413; ie 27.8%) appeared to be common to the CC-HEK-1 vs CC-HEK-2 based screening. This proportion is quite lower than the proportion of shared ORFs which are integrated (5005/6713; ie 74.6 %). This raises the question of sensitivity of the assay, that thus not seem to be led at “saturation”. This suggests there is an important proportion of false negatives in the screening, and that these false negatives differ from one experiment to another. Thus, this questions the comparability between the interactomes, which is put forward as one important asset of this new methodology.

Importantly, the interactomes of the WT HOXA9 and mutant HOXA9W (performed both with both cell lines) show an overlap which is similar in proportion: 23%. The differences between these interactomes might simply be explained by the low sensitivity of the assays and an important proportion of false negatives in each experiment. The authors indicate (lines 333-334) “several functions (…) were lost in the HOXA9W interactome”, but this could simply be explained by the fact that interactions were missed (experimentally) and not lost (intrinsically). Elsewhere the authors elaborate on the “ectopic functions revealed with the Trp mutation”, again, some interactions observed with the mutant might have been missed with the WT.

The sensitivity of the assay should be challenged by the reproducibility of its outcome. Ideally the authors should present several (N=3 or more) independent screenings run with the same protein (ie HOXA9 WT) on the same cell line (as suggested by the authors, lines 488-489) to conclude about the sensitivity of the assay and the comparability of the interactomes thereof.

Lines 486-489, while addressing this issue of reproducibility of the assay between the two cell lines, the authors suggest that “the frequency (…) and expression levels (depending on the genomic insertion site) between the two cell lines could influence the final enrichment (…)”. This is a possibility but the authors indicate that (1) to prepare the cell lines they reached (lines 114-115) “approximately one-gene-one-cell (…) with >500x representation” and next (2) that for the screen (lines 129-130) “8.106 CC-HEK-cells were thawed (~800x representation)”. This would imply that each construct should be represented by several independent integration events and each integration possibly represented by several cells which would preclude important integration loci effects: if some loci are not or poorly permissive for ORF expression, it should be excluded that most or all loci for a given ORF would be problematic.

Minor points

Lines 525-526 (and elsewhere in the manuscript) the authors highlight that an important proportion of interactors retrieved are not transcription factors. There might be a bias against retrieving TFs. The interaction between TFs might require their binding to DNA. The ability of VN-HOX and CC-ORFs (for example CC-PBX) to bind DNA (together) and to activate transcription should be tested. If the fusion proteins do not behave in transcription like wild-type ones, this might explain the relative paucity of TFs in the interactomes.

The authors investigated further one interaction involving HOXB13. To address the possible functional consequence of the HOXB13-CREB3L4 interaction, the authors invalidated the expression of the proteins by siRNA-mediated silencing. They recorded cell parameters with xCELLigence system, and both siRNAs resulted in similar moderate effects. When combining both siRNAs, the effects were more pronounced. From this the authors conclude “that HOXB13 and CREB3L4 could work as a cooperative dimeric complex”. This is indeed a possibility. Another possibility to discuss is that both proteins could act independently on the same cell processes, independently from their capacity to interact with each other. Their experiment did not address the role and importance of the HOXB13-CREB3L4 protein-protein interaction. It addressed the direct or indirect functional -not molecular- interaction between the HOXB13 and CREB3L4 ORFs.

Author Response

Reviewer 1

This manuscript by Yunlong Jia et al. presents a new methodology, they called Cell-PCA, to carry out proteome-wide protein-protein interaction screenings in live animal or human cell models. This assay is basically based on the BiFC concept, but it involves cells stably transfected for wide libraries of ORF-half-Venus fusions, which allows to use libraries for several assays to be run in parallel in order to lead to comparative interactomics in the same live cell context.

This methodology is innovative and clearly presented. The authors also provide convincing proof-of-concept experiments. A key issue in the approach however consists in the specificity and sensitivity of the Cell-PCA approach. The authors indicate that the specificity and sensitivity of Cell-PCA was established by comparing the interactomes obtained with wildtype and mutant HOXA9 proteins, but the outcome of these interaction screenings raise important questions mainly about the sensitivity of the assay. These questions should be addressed or discussed before acceptation of the manuscript for publication.

Major issue

The main question raised by the interpretation of the data presented in the manuscript concerns the sensitivity of the assay, which by essence renders questionable the comparability of the interactomes established by Cell-PCA.

The authors established two similar cell lines, CC-HEK-1 and CC-HEK-2, which were independently infected by two batches of the same library of lentiviral constructs coding for CC-ORFs. The characterization of these cell lines highlighted a high overlap of integrated constructs between the cells: 5005 integrated ORFs are shared by the two cell lines, 740 (CC-HEK-1) and 968 (CC-HEK-2) constructs being unique to either cell line. In contrast, while comparing the positive interactions retrieved with the same protein (HOXA9 WT) only 115 interactions (out a total of 413; ie 27.8%) appeared to be common to the CC-HEK-1 vs CC-HEK-2 based screening. This proportion is quite lower than the proportion of shared ORFs which are integrated (5005/6713; ie 74.6 %). This raises the question of sensitivity of the assay, that thus not seem to be led at “saturation”. This suggests there is an important proportion of false negatives in the screening, and that these false negatives differ from one experiment to another. Thus, this questions the comparability between the interactomes, which is put forward as one important asset of this new methodology.

We were surprised by this comment since it is usually the potential number of false positives that is criticized when performing BiFC experiments. In any case, we reconsidered our data in light of this comment. The values for the proportion of common interactions are comprised in between 20% and 40% (Fig. S9). These values have been changed compared to the first version, since we realized that we did not apply the log2FC>6 when counting the unique interactions. We discussed the different criteria that could explain part of these potential false negatives, although this proportion is in the range of other applications. For example, using a lower threshold of selection (log2FC>0,8) will significantly increase the number of positive ORFs. Still, we consider that it is more important/critical to don’t have false positives than false negatives, especially with our approach that gave more candidates when compared to other alternative approaches. This point is now mentioned in the discussion section: “This score is in the range of the reproducibility rate described for approaches based on high-throughput mass-spectrometry protein complex identification (around 19% when considering proteins present in two datasets: (von Mering C. et al., 2002) or Y2H (around 20%, (Brückner A. et al., 2009)). This proportion is however lower than the proportion of shared ORFs which are integrated (5005/6713; i.e., 75%), suggesting that there could be a proportion of false negatives. This proportion could be explained by our stringent log2FC criteria, which aimed at getting positive candidates with high confidence. It could also be explained by the fact that each CC-ORF was randomly inserted at a variable frequency in the genome (Fig. S4). Along this line, we noticed an inverse correlation between the number of common positive CC-ORFs and the variation of the insertion frequency score (based on the number of counts) between the two CC-HEK cell lines (Fig. S4). For example, the proportion of common positive ORFs between CC-HEK 1 and CC-HEK2 reaches 73% for HOXA1 when considering ORFs that vary less than 3 times between the two cell lines (Fig. S4). This point suggests that the level of reproducibility will probably be higher between replicates performed with the same CC-HEK cell line. Using CRISPR-based system for targeting a unique genomic insertion site for all CC-ORF constructs could constitute an interesting alternative in the future with this regard (Chi, X. et al., 2019).  Nevertheless, the overall number of positive interactions found in our screens was higher than the number of interactions obtained with other experimental approaches (see below), suggesting that the potential number of false negatives was not a strong limitation. On the contrary, given the high sensitivity of BiFC, it is particularly important to apply a high selection criterion to rather limit the number of potentially false positive interactions”.

Importantly, the interactomes of the WT HOXA9 and mutant HOXA9W (performed both with both cell lines) show an overlap which is similar in proportion: 23%. The differences between these interactomes might simply be explained by the low sensitivity of the assays and an important proportion of false negatives in each experiment. The authors indicate (lines 333-334) “several functions (…) were lost in the HOXA9W interactome”, but this could simply be explained by the fact that interactions were missed (experimentally) and not lost (intrinsically). Elsewhere the authors elaborate on the “ectopic functions revealed with the Trp mutation”, again, some interactions observed with the mutant might have been missed with the WT.

The similar proportion between HOXA9 and HOXA9w is pure coincidence (we have different proportions with other HOX). As mentioned previously, given the number of positive interactions (between 5% and 7% of the tested CC-ORFeome) we do not believe that the absence or gain of interactions for different HOX is mostly due to the presence of false negatives (or false positives). It was expected to lose interactions with this mutation. And the gain of interactions was also observed in previous BiFC screens with Drosophila Hox proteins, which is now mentioned in the manuscript.

The sensitivity of the assay should be challenged by the reproducibility of its outcome. Ideally the authors should present several (N=3 or more) independent screenings run with the same protein (ie HOXA9 WT) on the same cell line (as suggested by the authors, lines 488-489) to conclude about the sensitivity of the assay and the comparability of the interactomes thereof.

We fully agree with the Reviewer, and it is how it should be done in the future.

Lines 486-489, while addressing this issue of reproducibility of the assay between the two cell lines, the authors suggest that “the frequency (…) and expression levels (depending on the genomic insertion site) between the two cell lines could influence the final enrichment (…)”. This is a possibility but the authors indicate that (1) to prepare the cell lines they reached (lines 114-115) “approximately one-gene-one-cell (…) with >500x representation” and next (2) that for the screen (lines 129-130) “8.106 CC-HEK-cells were thawed (~800x representation)”. This would imply that each construct should be represented by several independent integration events and each integration possibly represented by several cells which would preclude important integration loci effects: if some loci are not or poorly permissive for ORF expression, it should be excluded that most or all loci for a given ORF would be problematic.

We agree with the Reviewer and we deleted this part. We now present more clearly the frequency representation of the 5005 genes present in the two cell lines (new Fig. S4A). We also present the proportion of common positive genes depending on their similar or dissimilar frequency insertion rate between the two cell lines. This analysis shows that the proportion of common positive genes is high when the frequency score is similar between the two cell lines (below 3x fold change). The proportion of common positives diminishes when the frequency insertion score differs more between the two cell lines (Fig. S4B). This observation highlights that our approach displays a very good level of reproducibility when considering genes with similar frequency insertion scores in the CC-HEK1 and CC-HEK2 cell lines. This point is now mentioned in the discussion section : “It could also be explained by the fact that each CC-ORF was randomly inserted at a variable frequency in the genome (Fig. S4). Along this line, we noticed an inverse correlation between the number of common positive CC-ORFs and the variation of the insertion frequency score (based on the number of counts) between the two CC-HEK cell lines (Fig. S4). For example, the proportion of common positive ORFs between CC-HEK 1 and CC-HEK2 reaches 73% for HOXA1 when considering ORFs that vary less than 3 times between the two cell lines (Fig. S4). This point suggests that the level of reproducibility will probably be higher between replicates performed with the same CC-HEK cell line.

Minor points

Lines 525-526 (and elsewhere in the manuscript) the authors highlight that an important proportion of interactors retrieved are not transcription factors. There might be a bias against retrieving TFs. The interaction between TFs might require their binding to DNA. The ability of VN-HOX and CC-ORFs (for example CC-PBX) to bind DNA (together) and to activate transcription should be tested. If the fusion proteins do not behave in transcription like wild-type ones, this might explain the relative paucity of TFs in the interactomes.

We agree it is an important point and we now mentioned our previous work that showed that the fusion topologies used for the HOX bait proteins did not affect DNA-binding or interaction with the generic PBX and MEIS cofactors: “In particular, this previous work established that the VN-HOXA9 fusion topology was appropriate for deciphering interaction properties with PBX1 and MEIS1 in vitro and in live cells (Dard et al, 2019a). We therefore decided to use the same VN-HOX9 fusion protein for performing the large-scale BiFC interaction screen”. Also in the third paragraph of the results: “HOX proteins were fused to the VN fragment at their N-terminus, since this fusion topology has previously been described to be compatible for deciphering interaction properties with PBX1 and MEIS1 in vitro and in live cells (Dard et al, 2018a, 2019b)”. We also precise in the first paragraph of the results section that using the C-terminal fragment makes all fusion quite neutral for the ORFeome: “In addition, the small size of the C-terminal fragment (82 residues long) makes it more neutral when compared to the N-terminal fragment (173 residues long) for fusion protein constructs (Hudry et al, 2011).”

Moreover, we do not consider that there is a relative paucity of TFs in HOX interactomes. On the contrary, interactions with TFs are enriched when considering their proportion in the 6713 tested CC-ORFs. This information is now more clearly mentioned in the main text: “As expected, interactions with TFs were enriched (40% of all HOX interactions, 595/1491) although TFs represent 20% of the tested CC-ORFs (1342/6713).”

The authors investigated further one interaction involving HOXB13. To address the possible functional consequence of the HOXB13-CREB3L4 interaction, the authors invalidated the expression of the proteins by siRNA-mediated silencing. They recorded cell parameters with xCELLigence system, and both siRNAs resulted in similar moderate effects. When combining both siRNAs, the effects were more pronounced. From this the authors conclude “that HOXB13 and CREB3L4 could work as a cooperative dimeric complex”. This is indeed a possibility. Another possibility to discuss is that both proteins could act independently on the same cell processes, independently from their capacity to interact with each other. Their experiment did not address the role and importance of the HOXB13-CREB3L4 protein-protein interaction. It addressed the direct or indirect functional -not molecular- interaction between the HOXB13 and CREB3L4 ORFs.

We agree with the reviewer and now mention that “Altogether, these results show that CREB3L4 could work as a collaborative partner of HOXB13, potentially through direct protein-protein interactions, for promoting its proliferative activity in prostate-cancer derived PC-3 cells.”

Reviewer 2 Report

This manuscript describes a new version of BiFC that uses a stably transfected cellular pool containing the preys, and a transient expression of the bait. As the authors illustrate, this system can be very useful to compare proteins. It also has potential to compare the dynamic of the interactomes upon cellular changes such as cellular cycles, stress or differentiation processes. The finding that TFs are mainly non-specific to HOXs but that the non-TFs are is interesting and novel.

This is a clever system and a well-written manuscript. 

As always in live-cell BiFC assays, there are inherent limitations to the system. As the authors study TFs that are crucial to development, is the method transposable in a cellular model that is more relevant, such as stem cells?

For this specific technique:

-how do we know which ORFs were not expressed and can this be factored in?

-What are the levels of expression compared to endogenous?

-When expressing the VN_baits, how is expression controlled in the different cells to identify false negatives? This is particularly important when comparing different HOXs.

l. 328- “The observation that 246 interactions were specific of HOXA9w also highlight that the Trp mutation induced more a gain than a loss of the HOXA9 interaction potential.” Is there independent evidence for a gain of toxic function from this mutant?

l.533- "Accordingly, HOX interactomes related to transcriptional regulatory processes contained distinct combinations of a majority of non-specific interacting TFs (Fig. S6). This observation underlines that HOX transcriptional specificity mostly relies on the establishment of specific combinations of interactions with TFs that have the potential to interact with several HOX proteins." Although not surprising, this is always interesting. Fig S^ doesn't really do justice to this statement. It would be nice to find an alternative analysis/ representation and include this in the main figures.

Author Response

Reviewer 2

This manuscript describes a new version of BiFC that uses a stably transfected cellular pool containing the preys, and a transient expression of the bait. As the authors illustrate, this system can be very useful to compare proteins. It also has potential to compare the dynamic of the interactomes upon cellular changes such as cellular cycles, stress or differentiation processes. The finding that TFs are mainly non-specific to HOXs but that the non-TFs are is interesting and novel.

This is a clever system and a well-written manuscript. 

As always in live-cell BiFC assays, there are inherent limitations to the system. As the authors study TFs that are crucial to development, is the method transposable in a cellular model that is more relevant, such as stem cells?

Yes the system is relevant for other cell types, and we now mention this point in the discussion section: In principle, this strategy can be applied in any cell line of interest as long as it is easily transfectable for getting a majority of VN-bait expressing cells. This parameter is important when considering that only a small proportion of these expressing cells will be positive for the cell sorting and subsequent NGS analysis.”

For this specific technique:

-how do we know which ORFs were not expressed and can this be factored in?

We now provided two additional supplementary tables: Table S3 (absent genes) and Table S4 (common integrated genes).

-What are the levels of expression compared to endogenous?

Although it is an interesting and important point, we have no cues about it. It is also for this reason that we voluntary kept a very low expression level for each CC-ORF (based on the leakiness of the TRE promoter, without Dox induction). Expression level of each CC-ORF should be addressed individually (and could change from cell to cell depending on the insertion site) and compared to the corresponding endogenous gene expression. As we discussed, the best system in the future will be to clone all CC-ORF on the same genomic landing site.

-When expressing the VN_baits, how is expression controlled in the different cells to identify false negatives? This is particularly important when comparing different HOXs.

Transfections of VN-HOX fusion proteins have previously been performed for quantifying and dissecting HOX/PBX/MEIS interaction properties in HEK cells and we followed exactly the same protocol for the BiFC screens. This previous work is now cited more clearly in the manuscript. It relies on the constitutive activity of the CMV promoter, with no expression differences between the different VN-Hox constructs under our experimental conditions.

  1. 328- “The observation that 246 interactions were specific of HOXA9w also highlight that the Trp mutation induced more a gain than a loss of the HOXA9 interaction potential.” Is there independent evidence for a gain of toxic function from this mutant?

This is indeed an interesting point and we discussed it a bit further in the corresponding result section: “These results recall previous observations with Drosophila Hox proteins, which have been shown to establish ectopic interactions and perform additional functions when mutated in the same Trp-containing motif (Merabet et al, 2007; Baëza et al, 2015)”.

l.533- "Accordingly, HOX interactomes related to transcriptional regulatory processes contained distinct combinations of a majority of non-specific interacting TFs (Fig. S6). This observation underlines that HOX transcriptional specificity mostly relies on the establishment of specific combinations of interactions with TFs that have the potential to interact with several HOX proteins." Although not surprising, this is always interesting. Fig S^ doesn't really do justice to this statement. It would be nice to find an alternative analysis/ representation and include this in the main figures.

We present a novel Fig.5 to better illustrate the different combinations of interactions with TFs in the context of the enriched RNA-polII function found with the different HOX proteins.

Round 2

Reviewer 1 Report

The authors improved the manuscript and took the main concerns raised upon the first review into consideration. In particular the point raised about the sensitivity of the method and the reproducibility between assays has been clarified and discussed in further detail. This allows the reader to better grasp the power as well as the limits of the approach.

The authors also modified their manuscript according to the few minor concerns raised upon the initial review.

In this revised form, the manuscript is now suitable for publication.